# Effects of Processing Methods and Inclusion Levels of Dried Garlic on In Vitro Fermentation and Methane Production in a Corn Silage-Based Substrate

**DOI:** 10.3390/ani13061003

**Published:** 2023-03-09

**Authors:** Juan Vargas, Federico Tarnonsky, Araceli Maderal, Ignacio Fernandez-Marenchino, Federico Podversich, Wilmer Cuervo, Camila Gomez-Lopez, Tessa Schulmeister, Nicolas DiLorenzo

**Affiliations:** Department of Animal Sciences, North Florida Research and Education Center, University of Florida, Marianna, FL 32443, USA; jvargasmartinez@ufl.edu (J.V.);

**Keywords:** essential oils, methane emissions, sulfur compound, supplementation

## Abstract

**Simple Summary:**

Herbs and spices have been used because they show antimicrobial properties. In this regard, garlic might modify rumen fermentation and potentially reduce methane. However, the effect of garlic inclusion on in vitro fermentation and methane production has been variable. The objective of this experiment was to evaluate the effect of different inclusion levels and processing methods of garlic on in vitro fermentation and methane production using a corn silage-based substrate. The initial condition of the garlic (intact or smashed), drying process (freeze-dried, oven-dried, or autoclaved), and garlic proportion in the diet (0, 2.5, and 5%) were evaluated in vitro. Thus, three different incubations were conducted, and in vitro fermentation and methane emissions were evaluated after 24 h of incubation. Neither the initial condition nor the drying process of garlic modified the fermentation and methane production. However, increasing garlic inclusions increased in vitro digestibility and the concentration of volatile fatty acids and ammonia and modified the proportion of acetate and propionate, suggesting greater in vitro fermentation. Methane was not affected when garlic was increased. Further work should be conducted in vivo to confirm the fermentation improvement when garlic is included in a corn silage-based diet.

**Abstract:**

Garlic (*Allium sativum*) contains secondary compounds that are known to modify rumen fermentation parameters and decrease methane (CH_4_) emissions. The objective was to evaluate the effects of increasing the inclusion levels and processing methods of garlic on in vitro fermentation and CH_4_ production. Treatments were arranged in a randomized complete block design with a 2 × 3 × 2 + 1 factorial arrangement, where the main factors were the initial condition of garlic (intact or smashed), drying process (freeze-dried, oven-dried, or autoclaved), and garlic proportion in the diet (2.5 and 5%) and one control (without garlic supplementation). Incubations were conducted using corn silage and cotton-gin trash (80:20, respectively) as basal substrates on three different days. Final pH, the concentration of volatile fatty acids (VFA) and ammonia nitrogen (NH_3_-N), in vitro organic matter digestibility (IVOMD), total gas production, and CH_4_ concentration were determined after 24 h. Initial garlic condition or drying processing neither modify (*p* > 0.05) the in vitro fermentation nor the CH_4_ production. However, increasing garlic inclusion linearly increased (*p* < 0.05) IVOMD, the concentration of the total VFA, and the proportion of propionate. Also, the concentration of NH_3_-N and the proportion of acetate increased quadratically (*p* < 0.05) with greater garlic inclusions. Finally, garlic inclusion did not affect (*p* > 0.05) gas and CH_4_ production. In conclusion, increasing garlic levels, but not the processing methods, improved in vitro fermentation but did not modify CH_4_ emissions under in vitro conditions.

## 1. Introduction

Cattle production has been criticized because of the greater use of natural resources and its emissions of greenhouse gases relative to other agricultural activities [1,2]. However, ruminants play an essential role in society, providing various ecosystem services and promoting a circular economy, especially in developing countries [3]. Ruminants can utilize by-products and fibrous feeds due to the extensive pre-gastric fermentation that provides energy and protein for the animal, resulting in high-quality food products such as milk or meat [4]. Indeed, non-edible human products constitute at least 72% of the ruminants’ diet under confined conditions and almost entirely in extensive grazing systems [5].

The global garlic industry produces and commercializes between 6.5 and 15.7 million tons of garlic, resulting in 3.5 million tons of garlic waste. Usually, garlic waste is discarded into landfills or burned, negatively affecting the environment [6]. Remediating food waste as animal feed is a viable option that has the potential to simultaneously address waste management, food security, and environmental challenges [7]. Raw garlic and garlic oil have been included in ruminants’ diets to modify the rumen microbial population and fermentation [8,9]. Garlic has sulfuric-containing compounds with antimicrobial, antiviral, and immunomodulatory properties [9,10,11,12]. Intact mesophyll garlic cells accumulate alliin; when garlic cells are disrupted due to mechanical damage, alliin is degraded to allicin. Allicin is an unstable molecule that is converted rapidly to diallyl tri, di, or mono-sulfide [10]. Sulfur-containing compounds inhibit the enzyme 3-hydroxy-3methylglutaryl coenzyme A reductase, affecting the synthesis of isoprenoid ethers, an essential compound of the archaeal cell membrane, resulting in lower methane (CH_4_) production [8,10]. However, garlic supplementation has shown contradictory results on the fermentation and CH_4_ production in vitro [8,13,14]. Different concentrations of secondary compounds, inclusion levels, processing methods, diet characteristics, and experimental conditions may explain this variability.

The concentration of secondary garlic compounds varies according to the garlic variety, agronomic practices such as sulfur fertilization, and garlic processing methods and storage conditions [10,11,15]. Garlic processing methods are expected to result in different proportions and concentrations of sulfur-containing compounds, modifying the biological and antimicrobial activity [15,16] and possibly affecting the fermentation dynamic differently. Additionally, garlic proportions and diet characteristics may modify rumen fermentation, CH_4_ production, and animal performance [10]. A greater concentration of sulfur-containing compounds is expected to increase the antimicrobial and antimethanogenic effect when increasing garlic inclusion [17]. Further, sulfur-containing compounds show greater permeability through the microbial membrane at a lower rumen pH. Thus, highly fermentable diets are expected to increase the permeability of the sulfur-containing compounds, resulting in a greater antimicrobial effect [18].

Although raw garlic has been included in high-concentrate diets [10], there is limited information regarding processing methods and increasing levels of raw garlic in vitro fermentation and CH_4_ production in backgrounding diets. Thus, this research aimed to evaluate the effects of different processing methods and increasing inclusion levels of raw garlic on in vitro fermentation and methane production in a corn silage-based diet. The experimental hypothesis was that freeze-dry and intact garlic and increasing garlic inclusion would modify the fermentation, reducing methane production in vitro.

## 2. Materials and Methods

### 2.1. Ethical Considerations

All animal procedures were approved by the University of Florida Institutional Animal Care and Use Committee (#202111460).

### 2.2. Location and Animal Adaptation

This experiment was conducted at the North Florida Research and Education Center in Marianna, FL. A total of two ruminally-cannulated Angus-crossbred steers (808.8 ± 36.3 kg of body weight) were used as ruminal fluid donors for the in vitro incubations. The steers were fed corn silage, cotton-gin trash, and a premix of a vitamins and minerals diet (700, 280, and 20 g/kg on a dry matter basis, respectively) at least 35 d before the collection of ruminal fluid.

### 2.3. Experimental Treatments

The diet that was fed to the steers during the adaptation period was used as a substrate for the in vitro incubations. Corn silage and cotton-gin trash were dried for 48 h at 55 °C. Corn silage and cotton-gin trash were ground to pass a 2 mm screen in a Wiley mill (Thomas Scientific, Swedesboro, NJ, USA) and analyzed for dry matter (DM), ash, crude protein (CP), neutral detergent fiber (NDF), and acid detergent fiber (ADF) at a commercial laboratory (Dairy One Laboratory, Ithaca, New York, USA, Table 1).

The treatments were designed to evaluate different processing methods and increasing levels of raw garlic in a backgrounding substrate composed of corn silage and cotton-gin trash mixture (70:30 on a dry matter basis, respectively). Accordingly, garlic cloves were maintained intact or smashed and then freeze-dried, oven-dried, or autoclaved. Then, garlic cloves were manually ground using a mortar. Further, processed garlic was included in two different proportions, 25 and 50 g/kg of the substrate substituting the corn silage and cotton-gin trash mixture. The experimental design was a randomized complete block design plus one control (without garlic addition) with a 2 × 3 × 2 + 1 factorial arrangement of the treatments, where the main factors were initial condition (intact or smashed), drying processing (freeze-dried, oven-dried, or autoclaved), and garlic proportion in the diet (25 and 50 g/kg).

### 2.4. Rumen Fluid Collection and In Vitro Incubations

Ruminal fluid, collected from a representative sample of digesta, was strained through four layers of cheesecloth, placed in pre-warmed thermos containers, and transported to the laboratory within 30 min of collection. In the laboratory, ruminal fluid was maintained under constant CO_2_ flux, combined in equal proportions from the two donor steers, and mixed with McDougall buffer to a 1:4 ratio of rumen fluid to buffer (i.e., inoculum). In vitro incubations were conducted on three separate days using the average of two bottles within incubation as a replicate.

The treatments were weighed into Ankom bags (0.70 g), heat sealed, and placed in a 125 mL serum bottle, with two bottles per treatment [19]. Briefly, inoculum (50 mL) was added to each bottle, including two bottles without substrate (blanks) under constant CO_2_ flux. The bottles were fitted with a butyl stopper, crimp sealed, and placed in an incubator for 24 h at 39 °C, set at 60 rpm. At the end of incubation, before removing the stopper, the final gas pressure was recorded using a manual transducer (Digital Test Gauge, Ashcroft Inc., Stratford, CT, USA), and a subsample of gas was collected from the bottle headspace. It was stored in vacuum vials to determine the concentration CH_4_. After removing the stopper, the final pH of fermentation fluid was recorded. There were two 10 mL subsamples that were collected, acidified by adding 100 µL of a 20% (*v/v*) H_2_SO_4_ solution to each subsample, and frozen at −20 °C until further analyses. The Ankom bags were removed from the bottles, washed with tap water until the effluent was clear, dried in a forced-air oven set at 60 °C for 48 h, and reserved until further analysis.

### 2.5. Laboratory Analysis

The concentration of VFA in the fermentation fluid samples was determined in liquid-liquid solvent extraction using ethyl acetate [20]. Briefly, the samples were centrifuged for 15 min at 10,000× *g*. Fermentation fluid supernatant was mixed with a meta-phosphoric acid (25% *w/v*): crotonic acid (2 g/L, internal standard) solution at a 5:1 ratio, and samples were frozen overnight, thawed, and centrifuged for 10 min at 10,000× *g*. The supernatant was transferred into glass tubes (12 mm × 75 mm; Fisherbrand; Thermo Fisher Scientific Inc., Waltham, MA, USA) and mixed with ethyl acetate in a 2:1 ratio of ethyl acetate to the supernatant. After shaking tubes vigorously and allowing the fractions to separate, the ethyl acetate fraction (top layer) was transferred to vials (9 mm; Fisherbrand; Thermo Fisher Scientific Inc., Waltham, MA, USA). The samples were analyzed by gas chromatography (Agilent 7820A GC, Agilent Technologies, Palo Alto, CA, USA) using a flame ionization detector and a capillary column (CP-WAX 58 FFAP 25 m × 0.53 mm, Varian CP7767, Varian Analytical Instruments, Walnut Creek, CA, USA). The column temperature was maintained at 110 °C, and injector and detector temperatures were 200 and 220 °C, respectively.

The concentration of ammonia nitrogen (NH_3_-N) was analyzed after centrifuging fermentation fluid samples at 10,000× *g* for 15 min at 4 °C (Avanti J-E, Beckman Coulter Inc., Palo Alto, CA, USA) following the phenol-hypochlorite technique [21] with the following modification: absorbance was read on 200 µL samples at OD620 in flat-bottom 96-well plates (Corning Costar 3361, Thermo Fisher Scientific Inc., Waltham, MA, USA) using a plate reader (Fisherbrand UV/VIS AccuSkan GO Spectrophotometer, Thermo Fisher Scientific Inc., Hampton, NH, USA).

The dried Ankom bags were ashed at 550 °C for 6 h to determine the undigested organic matter on the remaining fermentation residue. Thus, the in vitro organic matter digestibility (IVOMD) was calculated as shown:

IVOMD (%) = [(incubated organic matter − residual organic matter)/incubated organic matter] × 100

A gas subsample was analyzed to measure the CH_4_ concentration by gas chromatography (Agilent 7820A GC; Agilent Technologies, Santa Clara, CA, USA). A flame ionization detector was used with a capillary column (Plot Fused Silica 25 m × 0.32 mm, Coating Molsieve 5A, Varian CP7536; Varian Inc. Lake Forest, CA, USA). Injector, column, and detector temperatures were 80, 160, and 200 °C, respectively. The injector pressure was 20 psi with a total flow of 191.58 mL/min and a split flow of 185.52 mL/min with a 100:1 split ratio. The column pressure was 20 psi with a flow of 1.8552 mL/min. The detector makeup flow was 21.1 mL/min. The carrier gas was N_2_, and the run time was 3 min.

### 2.6. Statistical Analysis

The data were analyzed as a randomized complete block design with three replicates (incubation days) using the MIXED procedure of SAS version 9.4 (SAS Institute Inc., Cary, NC, USA). The average of the bottles within incubation day was considered the experimental unit, resulting in three replicates per treatment. The model included the fixed effects of the initial condition (intact or smashed), drying processing method (freeze-dried, oven-dried, or autoclaved), and garlic inclusion in the diet (2.5 and 5%) plus a control without garlic inclusion, and the random effect of incubation day (replicate). The means were compared using the Tukey test. Linear and quadratic effects due to increasing garlic levels were also determined. Significant differences were accepted if *p* < 0.05 and tendencies if *p* < 0.1. Main factors are reported and discussed because interactions between the main factors were not significant.

## 3. Results

### 3.1. Initial Condition and Drying Method of Garlic on In Vitro Fermentation and Gas Production

The fermentation fluid pH, concentration of total VFA and NH_3_-N, the proportion of VFA, and the production of gas and CH_4_ after 24 h of incubation were not different when the garlic was intact or smashed. However, the smashed garlic tended to increase (*p* < 0.1) the IVOMD (Table 2). On the other hand, the dried method of garlic did not modify (*p* > 0.05) fermentation fluid pH, the concentration of total VFA and NH_3_-N, the proportion of VFA, the IVOMD, and the production of gas and CH_4_ after 24 h of incubation (Table 2).

### 3.2. Increasing Garlic Proportion on In Vitro Fermentation and Gas Production

Increasing the inclusion of garlic linearly increased (*p* < 0.05) the fermentation fluid pH, the concentration of acetate and propionate, and the IVOMD (Table 2). Also, greater garlic inclusion tended to linearly increase (*p* < 0.01) the concentration of VFA and the proportion of propionate. Further, increasing garlic proportion showed a quadratic response (*p* < 0.05) on the concentration of NH_3_-N and tended to increase the proportion of acetate quadratically (Table 2). However, the concentration and the proportion of butyrate and the production of gas and CH_4_ were not different (*p* > 0.05) when increasing the proportion of garlic in the diet (Table 2).

## 4. Discussion

### 4.1. Initial Condition and Driying Method of Garlic on In Vitro Fermentation and Gas Production

Garlic supplementation has shown contradictory results on in vitro fermentation [13,22,23]. In this regard, different from the results that were observed in this experiment, intact and freeze-dried garlic was expected to preserve a greater concentration of active compounds (e.g., essential oils), potentially modifying the in vitro fermentation [15,16]. In this experiment, the initial condition of the garlic clove (i.e., intact or smashed) did not affect the in vitro fermentation, except that IVOMD tended to increase when garlic was smashed (Table 2). As mentioned before, intact and disrupted garlic cells present different compositions of sulfur-containing compounds [10], possibly affecting the microbial communities differently during fermentation [24]. Unfortunately, no reports have assessed the effect of intact or smashed garlic on rumen microbial communities. Then, it is possible that the secondary compound composition in the smashed garlic could modify the extent of the fermentation relative to the intact garlic. Additionally, it is expected that greater IVOMD results in a greater concentration of VFA or gas production [25]. In this regard, the greater observed variation in gas production and the concentration of VFA relative to the IVOMD could preclude the possibility of detecting differences between the intact and smashed garlic on those variables (Table 2).

On the other hand, the drying process did not change the fermentation dynamics in this experiment (Table 2), suggesting that the freeze-dry, oven-dry, or autoclaved process neither affected the chemical composition nor the presence of secondary compounds of garlic resulting in similar in vitro fermentation and gas production. Despite oven-drying may reduce the concentration of sulfur-containing compounds relative to freeze-drying [15], the low concentration of active compounds reported in garlic bulbs may possibly preclude recognizing differences among the drying methods [26]. Unfortunately, there are no reports of the effect of drying process of garlic on in vitro fermentation dynamics and future experiments using more concentrated products such as essential oils, should be conducted to prove this hypothesis.

There are no reports evaluating garlic’s drying process or initial condition on CH_4_ emissions. In this experiment, neither the initial condition nor freeze-drying changed the gas or CH_4_ production (Table 2). In this regard, the fermentation dynamic had little or no changes when including different processed garlic. Thus, it is not expected that the garlic process could decrease the CH_4_ emissions under in vitro conditions.

### 4.2. Increasing Garlic Proportion on In Vitro Fermentation and Gas Production

Increasing garlic inclusion has shown contradictory results on in vitro digestibility [13,22,27]. In this experiment, increasing garlic linearly increased the IVOMD (Table 2). Improving OM digestibility when increasing garlic inclusion is associated with changing the nutrient composition of the substrate because garlic shows less NDF concentration than corn silage and cotton gin trash [27,28]. Thus, corn silage and cotton-gin trash mixture substituted by increasing garlic levels resulted in a lower concentration of structural carbohydrates in the final substrate. In this regard, it is reported that structural carbohydrates are less degradable than soluble nutrients during the fermentation process, explaining the greater IVOMD when the garlic content was increased in the substrate [29]. Additionally, greater IVOMD was related to a linear increase in the concentration of total VFA, acetate, and propionate (Table 2), suggesting greater fermentability when garlic was included in the substrate [25].

In this experiment, the concentration of NH_3_-N showed a quadratic response when increasing garlic inclusion in the substrate, similar to other reports [23]. Garlic could increase the concentration of CP, resulting in greater NH_3_-N concentration [27]. Nevertheless, garlic supplementation affected proteolytic bacteria resulting in lower NH_3_-N [22,27]. Garlic inclusion possibly did not affect the microbial community at low supplementation levels (i.e., 2.5%) because the low concentration of antimicrobial compounds and increasing nitrogen in the substrate results in greater NH_3_-N concentration. However, higher garlic concentration (i.e., 5%) could affect proteolytic microbes by reducing the NH_3_-N.

Garlic inclusion modifies in vitro fermentation, reducing the acetate and increasing the propionate proportion [8], possibly due to changes in the rumen microbial population [12]. In this experiment, acetate and propionate proportions showed a quadratic and linear response when increasing garlic inclusion, respectively (Table 2). Similar to the concentration of NH_3_-N, fermentation may be affected differently according to the garlic inclusion level [30]. Additionally, the greater concentration of NH_3_-N at 2.5% of garlic inclusion could explain the greater proportion of acetate because acetogenic bacteria mainly utilize NH_3_-N as a source of nitrogen [29]. The acetate proportion was also reduced when the NH_3_-N concentration decreased in the fermentation.

Garlic inclusion reduced [18,23,31] or did not change [9] CH_4_ production in vitro. In this experiment, increasing garlic inclusion did not change the gas and CH_4_ production. Garlic shows sulfur-containing compounds that are related to antimicrobial and antimethanogenic effects [9,10,11]. However, the presence of secondary compounds varies according to garlic variety, agronomic practices, and environmental conditions [11]. Variations in VFA proportion and an increased in IVOMD suggested that increasing garlic inclusion modify the ruminal microbial community and fermentation dynamic; however, the inclusion of garlic did not affect the CH_4_ synthesis suggesting that the garlic effect is more related to other microbial communities that are different from the methanogenic, or some methanogenic communities could overcome the effect of garlic supplementation. Further research should be conducted to prove those hypotheses.

## 5. Conclusions

Processing methods of raw garlic did not modify in vitro fermentation and CH_4_ emissions in a corn silage-base substrate. However, increasing the raw garlic inclusion changed the concentration of NH_3_-N and the proportion of acetate and propionate, suggesting different fermentation mechanisms or changes in the microbial community according to the garlic inclusion.

## Figures and Tables

**Table 1 animals-13-01003-t001:** Chemical composition ^a^ (g/kg of the DM) of the ingredients that were used in the experiment.

Item	Corn Silage	Cotton-Gin Trash	Raw Garlic
Organic matter ^b^	972	897	950
Neutral detergent fiber	274	617	62
Acid detergent fiber	166	588	49
Crude protein	80	160	183
Ether extract	34	32	49
Ash	28	103	50

^a^ Determined in a commercial laboratory (Dairy One Laboratory, Ithaca, New York). ^b^ Organic matter = 100 − %Ash.

**Table 2 animals-13-01003-t002:** Processing method and increasing proportion of garlic on fluid fermentation pH, the concentrations of volatile fatty acids and ammonia-N, in vitro organic matter digestibility, and gas and methane production in a corn silage-based diet.

Variable	Control	Freeze-Dry				Oven-Dry				Autoclaved				SEM ^1^	*p*-Value ^2^		
		Intact		Smash		Intact		Smash		Intact		Smash			Dry	IC	Inc
		2.5	5	2.5	5	2.5	5	2.5	5	2.5	5	2.5	5				
pH	6.59	6.61	6.70	6.64	6.67	6.69	6.67	6.67	6.66	6.67	6.67	6.69	6.63	0.050	0.932	0.796	0.007 L
Total VFA, m*M*	59.29	64.00	59.09	61.96	63.52	60.63	62.42	63.17	63.30	63.02	61.73	60.61	63.77	5.970	0.993	0.588	0.053 L
Acetate, m*M*	23.97	26.39	25.69	25.32	25.89	24.58	25.41	26.05	25.71	25.75	24.78	25.45	25.53	2.621	0.821	0.724	0.004 L
Propionate, m*M*	21.66	23.62	22.07	22.20	24.09	21.40	24.26	23.54	24.58	23.81	21.06	22.49	23.40	2.051	0.754	0.804	0.037 L
Butyrate, m*M*	5.47	5.65	5.80	5.79	5.48	5.80	5.16	5.59	5.32	5.51	5.71	5.79	5.94	0.381	0.453	0.802	0.357
Acetate, mol/100 mol	40.39	40.85	41.14	40.92	40.69	40.56	40.67	41.18	40.50	40.82	37.97	41.98	39.98	0.869	0.886	0.582	0.079 Q
Propionate, mol/100 mol	36.41	36.91	35.77	35.95	38.17	35.57	39.02	37.35	38.94	37.92	36.66	35.05	36.69	1.601	0.343	0.945	0.090L
Butyrate, mol/100 mol	9.33	8.95	9.45	9.41	8.65	9.59	8.33	8.91	8.54	8.81	9.32	9.55	9.44	0.506	0.312	0.957	0.153
Acetate:Propionate	1.12	1.11	1.16	1.15	1.07	1.15	1.05	1.11	1.05	1.08	1.10	1.21	1.09	0.066	0.450	0.785	0.054
Ammonia-N, m*M*	15.55	17.17	13.37	17.56	15.16	16.13	15.43	16.57	15.38	16.62	13.66	16.28	16.79	1.659	0.997	0.239	0.023 Q
IVOMD ^3^, %	56.01	57.87	55.45	58.48	59.38	55.83	57.84	59.77	59.83	58.53	58.56	58.45	59.74	3.123	0.923	0.071	0.013 L
Gas production, mL/gOMd ^4^	100.19	99.59	98.24	98.31	97.14	94.10	93.20	97.65	104.56	99.26	90.74	92.49	99.07	7.195	0.638	0.364	0.231
Methane production, m*M*/gOMd ^4^	0.42	0.43	0.39	0.42	0.42	0.40	0.46	0.46	0.40	0.38	0.38	0.41	0.43	0.077	0.730	0.505	0.951

^1^. SEM: Standard error of the mean, *n* = 3. ^2^. Dry: Effect of dry treatment (freeze-dry, oven-dry, or autoclaved). IC: Effect of the initial condition of garlic (smashed or intact). Inc: Effect of garlic inclusion (25 and 50 g/kg of the diet). L: Lineal effect. Q: Quadratic effect. ^3^. IVOMD: In vitro organic matter degradability. ^4^. OMd: Organic matter degraded.

## Data Availability

The data that are presented in this paper are available on request from the corresponding author.

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
