# Peer review of "Effects of Processing Methods and Inclusion Levels of Dried Garlic on In Vitro Fermentation and Methane Production in a Corn Silage-Based Substrate"

_animals, 2023, doi:10.3390/ani13061003_

Round 1

Reviewer 1 Report

The description of the methods is very confused. The word diets is used both for the two fed to the two steers and the 18 used for the in vitro studies. Use (e.g.) substrates or media for the latter.

The results and discussion should be separated. The numbers and lack of significance of differences need to be described in more detail before embarking on any possible explanations.

In essence, the paper reports no effects of inclusion of garlic wastes on the in vitro fermentations. This could well be because the rumen liquor samples were taken from only two steers fed identical diets and therefore the rumen microbiota used for all in vitro studies was similar.  The proper (more expensive ) experiment would be to study in vitro fermentation patterns with rumen microbiota long-term adapted to diets with different levels of incorporation of garlic residues.

Author Response

The authors thank the reviewer comments. We consider appropriate to improve the document.

  1. According to the reviewer’s comments, I modified the methodology to clarify it. Also, I changed the word diet by substrate, referring to the mixture used during the in vitro incubations.
  2. According to the reviewer’s comments, I divided the results and discussion into two different sections. Additionally, I improved the discussion by mentioning the absence of differences among the treatments and the possible explanations.
  3. I understand the reviewer’s comment. However, I do not agree with that perception for two reasons. First, the reviewer mentioned the cost and logistics of adapting the rumen microbial community to garlic supplementation for 3 to 5 weeks previously to collecting rumen fluid for incubation. The second reason is that the rumen can adapt to external factors degrading or producing non-toxic metabolites and restoring the initial condition after the adaptation period. For example, the addition of ionophore in ruminant diets decreased methane emissions for 2 weeks, but methane emissions were restored after the adaptation period. For this reason, I consider the first step to evaluate an additive should be using a non-adapted rumen microbial to recognize this initial effect on the in vitro fermentation. Differently, if the additive requires the adaptation of a specific community for metabolizing it or avoid animal toxicity, for example, nitrates, that require increasing a microbial population to avoid the NO2 accumulation and potential animal health problems.

Reviewer 2 Report

Overall, this manuscript is written well, and the sections follow clearly.

Materials and Methods are generally appropriate, although the experimental approach, sampling, and statistical analysis parts are confusing, and a few details should be clarified.

Please note that page numbers are missing for a couple of pages (table and conclusion pages), and the references pages have different page orders (2 of 9 and 3 of 9).

Due to this, I made my comments on the table and conclusion pages as #6 and  #7.

Specific comments follow:

Abstract:

p. 1, lines 38-39: The P-value is missing in Table 2.

Materials and Methods:

p. 3, lines 100 – 101: What was the point of using two steers, and how did authors handle the sampling variation from two steers?

Please distinguish biological and technical replicates.

p. 3, lines 120 – 123: more detail here is required.

Were there different proportions included on the same day?

p. 3, lines 130: more detail here on sampling is needed. On three separate days from the same steer or three separate days for three different garlic proportions?

Replicates refer to what?

p. 3 – 4, lines 133 – 134: Were samples from 2 steers pooled or processed separately?

Statistical analysis:

p. 5, lines 193: Please include the SAS version.

p. 5, lines 194 - 197: Clarification on biological and technical replicates is required.

It is unclear whether samples from 2 steers were pooled and how they were analyzed.

Table 2:

p. 6, Table 2: The table is incomplete, and the last 2-3 columns are missing.

Does P2 refer to the P-value? If yes, please clarify it in the table.

Does n=3  refer to biological or technical replicates?

Last page:

p. 7, lines 278 - 230: P-values are missing in Table 2.

p. 7, lines 230 – 231 and 242 – 244: Results in Table 2 do not support this statement. Authors should provide P-values.

Author Response

The authors thank the reviewer's comments. We consider it appropriate to improve the document.

  1. Throughout the materials and methods sections, more detail was included to clarify the procedure, as the reviewer suggested.
  2. The number of pages was corrected, as the reviewer noticed.
  3. P-Value was included in the abstract as the reviewer suggested.
  4. Usually, two or more steers are used to taking account the microbial variation. In this case, we mixed the rumen fluid to homogenize the microbial community but incorporate the natural variation presented between animals.
  5. In this experiment, incubation day is the replicate. Bottles within the day are the pseudo-replicate. In this regard, we used the average of the two bottles on the incubation day as the experimental unit.
  6. The relationship between rumen fluid and buffer was always the same.
  7. SAS version was included as the reviewer suggested.
  8. Table two was modified to incorporate the reviewer´s suggestions.
  9. The result and discussion sections were modified to improve the clarity and fluidity of the reading, as the reviewers suggested.

Reviewer 3 Report

Overall the paper needs substantial revision prior to publication

Simple Abstract: L21 and L25: Effect on methane emissions repeated. 

L36-40: The lack of effects on methane need to be mentioned as this was a clear research objective 

Introduction:

L45: Greater use of natural resources? Compared to what?

L51-52: Constituted or constitute? In what production system? TMR based?

L63-70: This paragraph seems out of place? As the paragraphs before and after are discussing the effects of garlic on methane and fermentation

L81: high to highly. Also this sentence needs further expansion 

I am a little confused by the introduction. A large portion of the introduction focuses on garlic waste but it is not clear how this applies to the paper? The introduction needs to be more focused on what makes this work novel. There is little logic introduced for the processing of the garlic that is a large portion of the study 

L137: It is not clear what the experimental unit is? Were two bags placed in each vessel? How many vessel per treatment? 

L144: Were the vessels opened to collect the sample of methane? What were the volume of gas collected? 

L197: Again it is unclear what the number of replicates used is 

L207-210: This was not measured? How do you know that the lack of effect was related to a potential difference in oil content? The lack of  composition analysis of the included garlic is a major issue with the manuscript 

Table2: The full table is not visible 

Also what does the control refer to? It is not clear from the M&M

The results/discussion is very difficult to follow and needs substantial revision 

Author Response

The authors thank the reviewer's comments. We consider it appropriate to improve the document.

  1. Into the simple abstract “Effect on methane emissions repeated” sentence is not repeated because it refers to different factors. Initially, it refers to the effect of garlic processing and then to garlic inclusion.
  2. According to the reviewer’s comment, methane was included in the objective.
  3. According to the reviewer’s comment, the expression “great” was changed, resulting in clearer text.
  4. According to the reviewer’s comment, the word “constituted” was changed to “constitute” and the sentence was rewritten for better comprehension.
  5. According to the reviewer’s comment, the introduction was modified to improve the comprehension of the text. Also, there was an effort to highlight the novelty of the experiment.
  6. The experimental unit is the bottle that contains one Ankom bag. In this regard, the material and methods section was reviewed to make it clear. Additionally, the number of replicates is explained in the statistical analysis section.
  7. Gas was collected and stored in pre-vacuum vials and analyze for methane concentrations. In this regard, this point was addressed in the materials and method section.
  8. The control is defined as the substrate without garlic.
  9. According to the reviewer’s comment, the discussion section was substantially improved according to the experimental results.
  10. I can visualize the entire table 2. I do not understand what the reviewers refer

Round 2

Reviewer 1 Report

The presentation has been improved.

It is right and proper that negative results should be published. However, I fail to be convinced that with this experimental design you could have expected anything else.

Author Response

We thank the referee for your comments. We consider them appropriate to improve this document.

We agree with the reviewer regarding the results of this experiment. We consider it an exploratory investigation to test if the garlic process or inclusion could modify the in vitro fermentation. Raw garlic does not change in vitro fermentation; different approaches should be evaluated as essential oils. It requires incorporating other technology (e.g., oil extraction) and possibly will need greater investment to be incorporated in farms. However, not much information regarding using raw garlic suggests valuable current experimental results.

Reviewer 3 Report

The manuscript while improved there is still some issues that need attention.

In the uploaded PDF I still am missing half of the text from the table, it may be fine in the word document but authors need to check what is uploaded.

L139: The number of replicates are still not clear. They should be outlined here also as well as the statistical section. 

L205: drying method not dried 

L206: Start the section with the NS results 

There is inconsistency in the presentation CH4 and CH4

Is the nutrient specification available for the garlic?

Author Response

We thank the referee for their comments. We consider them important to improve this paper.

The table is depicted correctly. We believe we can be sure about the final presentation during the edition process.

In this experiment, two bottles per treatment per incubation day were incubated with the same substrate (L143). The experimental unit is the average of two bottles with the incubation day (L199). And the incubation day were three different days (L140). In this regard, the number of replicates is three per treatment (i.e., the average of the two bottles x the number of incubations). We clarify the redaction in the materials and methods section.

Dried was changed by drying according to the reviewer's suggestion.

The order of the results was modified according to the reviewer's suggestion.

The subscripts of CH4 were checked and corrected adequately according to the reviewer's suggestion.

Unfortunately, garlic samples were utilized during the experiment and were not chemically analyzed.